# Relevance of Potential Contributing Factors for the Development and Maintenance of Irritability of Unknown Origin in Pediatric Palliative Care

**DOI:** 10.3390/children10111726

**Published:** 2023-10-24

**Authors:** Larissa Alice Kubek, Nina Angenendt, Carola Hasan, Boris Zernikow, Julia Wager

**Affiliations:** 1PedScience Research Institute, 45711 Datteln, Germany; b.zernikow@pedscience.de (B.Z.); j.wager@pedscience.de (J.W.); 2Department of Children’s Pain Therapy and Paediatric Palliative Care, Faculty of Health, School of Medicine, Witten/Herdecke University, 58448 Witten, Germany; nina.angenendt@uni-wh.de (N.A.); c.hasan@kinderklinik-datteln.de (C.H.); 3Paediatric Palliative Care Centre, Children’s and Adolescents’ Hospital, 45711 Datteln, Germany

**Keywords:** irritability, palliative care, pediatrics, children

## Abstract

Potential contributing factors (PCFs) for irritability of an unknown origin (IUO) in children with neurological conditions are identifiable through structured diagnostics. Uncertainty exists regarding the actual relevance of identified PCFs to IUO. Assessments from parents as well as nursing, psycho-social, and medical professionals were used to determine the contribution of different PCFs in the development and maintenance of IUO. For this, individual PCFs of N = 22 inpatient children with IUO were presented to four raters. Descriptive statistics, Kruskal–Wallis tests, and Krippendorff’s alpha were used to determine which PCFs were most relevant to explain IUO and rater agreement. Psycho-social aspects (44.7%), hyperarousal (47.2%), pain (24.6%), and dystonia (18.1%) were identified as the most relevant PCFs for IUO. Descriptively, physicians’ relevance rating regarding psycho-social aspects, hyperarousal, and dystonia deviated the most from the overall group rating. All professional raters considered psycho-social aspects to be more relevant than did parents. Parents rated pain as more relevant than the other raters. Kruskal–Wallis tests showed no significant differences between relevance ratings (H = 7.42, *p* = 0.059) or the four parties’ deviations (H = 3.32, *p* = 0.344). A direct comparison of the six two-party constellations showed that across all factors, agreement was weak to moderate. The highest agreement was between physicians and nurses (α = 0.70), and the lowest was between nurses and psycho-social experts (α = 0.61). Understanding which psycho-social and various biological PCFs are significant for IUO can facilitate more targeted and individualized pediatric palliative care for affected patients.

## 1. Introduction

Children with severe neurological conditions, who are under the care of pediatric palliative teams, often suffer from a highly unpleasant and complex “irritability of unknown origin (IUO)” [1,2,3,4,5]. According to the definition of Siden [6], a child with IUO shows distress behavior characterized by vocalizations and movements of the body, typically arching and stiffening, especially of the extensor muscles. This distress behavior occurs in episodes whereby the length of the episodes is unpredictable and can last from a few minutes to hours [7]. To make matters worse, these patients suffer from a high overall symptom burden and, in addition to IUO, have to cope with multiple other symptoms such as seizures or movement disorders [2]. Distressing comorbidities such as feeding, nutritional, or gastroenterological issues commonly add to the problem [8]. Further, a particular challenge in these patients is the frequent severe psychomotor impairment and non-verbality associated with the underlying disease, due to which they are unable to report their own sensations [6,9,10,11,12,13,14].

In the literature, IUO and pain are often not strictly distinct, and sometimes are even equated. This circumstance is understandable, given the indisputable similarity in how both conditions can manifest in an affected child [6,13,15]. However, there is clear evidence that even though pain is often identified as a contributing factor for IUO, IUO and pain are indeed distinct conditions that should be recognized and treated as such in clinical practice [6,7,13,15]. IUO can significantly reduce both the patient’s and the parents’ quality of life, for example, by disrupting or not establishing normal family routines, causing parental mental health problems, leaving siblings with insufficient attention, or forcing parents to terminate their employment relationship [16]. Because IUO is so stressful for patients, parents, and professional caregivers, the accurate diagnosis of potential contributing factors (PCFs) is essential. Nevertheless, parents in particular often report dissatisfaction with IUO diagnostics. This may be due to the feeling that too little has been carried out to get to the bottom of the exact causes of IUO, or the fact that too many burdensome and costly measures have been carried out [16]. One possible reason for this could be the hitherto absence of established approaches to the structured recording and treatment of IUO [16].

To counter this, a recently published paper presented a systematic approach to identify PCFs of IUO in the clinical setting through basic diagnostics, special diagnostics, and hypothesis-guided therapy [7]. This approach has demonstrated promise in detecting various biological and psycho-social PCFs. Consequently, it serves as a crucial foundation for managing a child’s IUO and contributes to a deeper understanding of the IOU’s etiology in general. A particularly pertinent finding is that PCFs occur cumulatively, which again underscores the high complexity of IUO. In addition, not only should biological PCFs be in the focus of diagnostics; psycho-social PCFs must urgently be accorded equal attention [7]. To further enhance our understanding of IUO and to tailor therapeutic approaches effectively and individualize them to specific PCFs, we need to clarify the extent to which each identified PCF contributes to a patient’s IUO. For example, a child might suffer from an acute urinary tract infection as a PCF, but an interaction problem with the parents may have the most impact on IUO.

Accordingly, the aim of this study was to determine whether or not, and to what extent, identified PCFs might be involved in IUO. To capture comprehensive data, the assessments of different pediatric palliative care professionals and parents of inpatient children with IUO were collected. In addition, we aimed to examine the extent to which the different groups of raters agreed among their judgments, as this aspect could provide significant diagnostic insights.

## 2. Materials and Methods

### 2.1. Setting

This study describes the second part of a prospective observational study on IUO, which was conducted in a German pediatric palliative care unit. The first part’s results were recently published [7].

Since 2010, the pediatric palliative care unit has provided services for about N = 170 patients facing a diverse range of life-limiting conditions annually and follows a multiprofessional, bio-psycho-social pathway. In this institution, affected children and adolescents can be cared for from the beginning of their conditions and are thus often readmitted several times over many years. In addition to IUO, seizures, pain, spasticity, and sleep disorders are common among young patients [17].

### 2.2. Participants

The study period spanned the years 2018–2020, during which time children attending the pediatric palliative care unit for IUO were recruited. To be eligible for study inclusion, IUO could present itself as an admission diagnosis, or develop during the inpatient stay. Additionally, informed parental consent was required for participation.

Patients were excluded from the study if (a) they were at the acute terminal stage of life, (b) the family was in an acute psycho-social or medical crisis, (c) the parents had insufficient local language proficiency, or if (d) the child expired during the inpatient stay. Ethical approval was obtained by the Ethics Committee of the Witten/Herdecke University (approval code: 145/2018, approval date: 10 November 2018). Informed consent was obtained from all families to participate in the study.

### 2.3. Data Collection

Following the diagnostic and therapeutic approach to IUO developed as a standard operating procedure in the pediatric palliative care unit, in the first phase of the study all IUO-related potential contributing factors (PCFs), the associated diagnostic actions leading to their identification, and the therapeutic outcomes of IUO were determined [7]. The standard diagnostic and therapeutic approach in the pediatric palliative care unit includes comprehensive basic diagnostics, special diagnostics, and hypothesis-guided therapy. Basic diagnostics include, for example, a medical, psycho-social, and nursing history and physical examinations. Patients’ medical records, assistive devices, and medication dosages are assessed. Any potential interactions and/or adverse drug reactions are checked and basic laboratory tests are initiated. If basic diagnostics do not detect a PCF of the IUO or if primary treatment does not result in adequate symptom improvement, special diagnostics are initiated. Special diagnostics are, for instance, radiological and sonographic and other assessments by physicians treating the patient. Which specific measure is initiated and in what order they are initiated depends on the professional judgment of the practitioner. Some suspected PCFs of IUO (e.g., visceral hyperalgesia) cannot be detected reliably in the absence of robust diagnostic measures. Attempts at hypothesis-guided therapy are made in these cases. IUO therapy is started, for example, when a child responds to hypothesis-guided therapy, as this could signal an IUO PCF. Therapeutic strategies may include cause-oriented or symptomatic medication treatment [7]. Overall, the diagnostic and therapeutic approach reflects many years of profound clinical experience and expertise from various professions (e.g., nursing, psycho-social, and medical) and is informed by the current state of scientific knowledge concerning IUO in pediatric patients with life-limiting conditions.

In collecting data for the first part of this study, all abnormalities (medical and psychosocial) that were identified in the course of routine diagnostics and specifically marked as such in the patient records were considered potentially contributing factors (PCFs) to IUO. These PCFs, along with the specific diagnostic steps taken to identify them and the associated therapeutic outcome, were then extracted from the patient files and entered daily into a case report form. Further details regarding the respective procedure can be found in the corresponding publication [7].

For the second part of the study, at the end of the inpatient stay, each child’s documented PCFs were presented individually by a member of the study team to one of the child’s parents and to one professional from each of the professions of medicine, nursing, and psycho-social work in the pediatric palliative care unit that cared for the child. Table 1 displays the 16 PCFs identified in the first part of the study that were presented to the raters [7].

These raters used their personal expertise to subsequently reassess the relevance of each PCF to the child’s IUO that had been addressed during inpatient care at the time. For this weighting, 100 percentage points in total could be allocated across all PCFs (example: if a child had pain, hypersecretion, and parental interaction problems documented as PCFs, the nurse could assign 40%, 5%, and 55%, respectively). No rater had insight into the other raters’ assessments. For some analyses, PCFs were divided into psycho-social and biological aspects. Biological aspects included all categories from Table 1 except psycho-social aspects.

### 2.4. Data Analysis

All analyses were conducted using the SPSS statistics software (IBM, version 28) and the R software environment for statistical computing (https://www.r-project.org/, accessed on 14 April 2022) with the “psych” package [18].

Patient characteristics were analyzed using descriptive statistics. Given the unchanged sample in study parts one and two, basic patient information is provided in this paper while specific details are given in the corresponding publication [7].

To determine the relevance of PCFs for IUOs at a global level, the average percentage points assigned to biological and psycho-social PCFs by the four rating parties were calculated for all children.

Next, the relevance of individual PCFs was derived from the mean value of all ratings for a given factor. To assess the extent to which the individual rating parties shared the overall impression of the group, first, the mean of the four parties’ ratings for each child with a corresponding PCF was calculated. Then, for each party, its average deviation from the patient-weighted means of the PCFs was established. For all mentioned analyses, potential group differences were analyzed using a Kruskal–Wallis test.

The direct differences between the judgments of the rating parties were determined for each of the six possible two-party constellations (parent vs. medicine, parent vs. psycho-social, parent vs. nursing, nursing vs. psycho-social, nursing vs. medicine, and psycho-social vs. medicine) descriptively and via Krippendorff’s alpha. For this, the following conventions were used for interpretation (maximum range 0–1): α > 0.8 (strong agreement), α = 0.67–0.8 (moderate agreement), and α < 0.67 (weak agreement; [19]).

By subtracting the ratings of the second party from the ratings of the first party, it was possible to not only ascertain the amount of deviation, but also which of the two parties considered a factor to be more relevant than the other (Figure 1).

## 3. Results

### 3.1. Patient Characteristics

On average, the N = 22 included patients (n = 14 male, n = 8 female) were 9 years old (range: 0–25 years) and all were non-verbal. Their inpatient stay duration averaged 20 days (range: 4–40 days). The three most common underlying diseases (according to the ICD-10) were P91, other disturbances of cerebral status of newborn (n = 7, 31.8%); Q87, other specified congenital malformation syndromes affecting multiple systems (n = 5, 22.7%); and E70-E90, metabolic disorders (n = 4, 18.2%). All data can be found in [7].

### 3.2. PCF Relevance

Across all children, parents gave an average of 67 percentage points to biological PCFs, nurses assigned 55.7 percentage points, psycho-social professionals gave 54.2 points, and medical professionals gave 53.2 points. Regarding psycho-social aspects, parents assigned an average of 33 points, nurses assigned 44.3 points, psycho-social professionals assigned 45.8 points, and medical professionals assigned 46.8 points. There were no significant differences between the ratings of the four parties, neither for the biological nor for the psycho-social factors (Kruskal–Wallis test; H = 4.18, *p* = 0.24).

Psycho-social aspects were considered the second most relevant PCF for the development and maintenance of IUO across all raters (44.7%, see Figure 2a). Among biological factors, hyperarousal (47.2%), pain (24.6%), and dystonia (18.13%) were identified as the three most relevant factors.

Descriptively, nurses allocated their highest average PCF rating to hyperarousal, as did medical professionals and parents. Psycho-social raters, on the other hand, considered psycho-social aspects to be the most relevant PCF for the development and maintenance of IUO. Breathing problems were initially suspected as a PCF for IUO in four children [7], but were ultimately rated as irrelevant by all raters when compared to the other factors (see Figure 2b). Across all individual PCFs, there was no significant difference between the four average ratings (Kruskal–Wallis test, H = 7.42, *p* = 0.059).

For the psycho-social PCFs, physicians deviated the most from the average overall group impression. This was also the case for hyperarousal and dystonia. For pain, parents were descriptively least consistent with the overall group’s judgment (see Table 2).

Across all factors, parents deviated the most from the overall group most frequently (Table 2). Nevertheless, a Kruskal–Wallis test showed no significant differences between the deviations of the four parties (H = 3.32, *p* = 0.344).

### 3.3. Direct Deviations between the Parties’ Relevance Ratings

A direct comparison of the six possible two-party constellations was made for the four most relevant PCFs (psycho-social aspects, hyperarousal, pain, and dystonia). Descriptively, parents’ and medical professionals’ ratings of psycho-social aspects were most different. Furthermore, there were large discrepancies between parents and the nurses and psycho-social professionals—on average, nearly 15 percentage points. Overall, all professional raters descriptively attributed greater relevance to psycho-social aspects for IUO than did parents (see Figure 3a).

Psycho-social and medical professionals agreed least on how relevant hyperarousal was to IUO. The second largest discrepancy was between parents and medical professionals. In both cases, the latter rated hyperarousal as more relevant to IUO than did the other parties (see Figure 3b).

Pain was the sole factor to which parents assigned a higher average relevance to IUO when directly compared with professional raters, showing, for example, an average 12 percentage point difference between parents and medical and psycho-social professionals (see Figure 3c).

For dystonia, the judgments made by nurses diverged most from those of medical professionals, and parents’ judgments differed most from those of medical professionals. In both cases, physicians rated dystonia as more relevant to IUO compared to the other parties. Parents and caregivers, on the other hand, gave similar judgments (Figure 3d).

Across all factors, Krippendorff’s alpha indicated weak to moderate agreement among the six rater constellations (range: 0.61 to 0.70). The highest agreement was observed between physicians and nurses, while the lowest agreement was between nurses and psycho-social professionals (Table 3).

## 4. Discussion

Part 1 of this study, utilizing a standardized diagnostic approach, demonstrated that numerous PCFs may account for the development and maintenance of IUO in pediatric palliative care [7]. Part 2 aimed to evaluate and quantify the true relevance of these PCFs to IUO based on the judgments of professionals and parents. Psycho-social aspects, hyperarousal, pain, and dystonia were identified as the most relevant PCFs for IUO.

This shows once more that IUO is not solely determined by biological phenomena, but that the psycho-social characteristics of the child and the family system should also be considered in potential therapeutic approaches for IUO.

The inclusion of pain in the list of most relevant factors, while others like hyperarousal or psycho-social aspects are deemed even more important, reinforces that IUO should not be equated with pain, contrary to some of the literature [6,7,13,15]. Instead, recognizing that pain can be one of several different causes of IUO simplifies practical approaches in daily clinical practice by creating a clear distinction between both phenomena, allowing for more tailored therapeutic approaches.

Considering all potential factors, hyperarousal was rated as the most relevant factor for IUO in the included children. This implies that central nervous system involvement in IUO is frequently implicated, and the successful treatment of other biological factors may not necessarily lead to improvement in IUO, despite their potential relevance. This again illustrates the highly complex nature of IUO [1,6,20] and underscores that despite extensive efforts, a complete resolution of IUO may not always be attainable, though symptom improvement can be achieved. Furthermore, it is possible that the raters in our study aimed to emphasize this circumstance by assigning high relevance to hyperarousal.

A noteworthy finding is that all professional raters considered psycho-social aspects to be more relevant to children’s IUO compared to parents. In the context of the study, professionals may be more sensitive to the importance of psycho-social factors, such as interaction problems or parental conflict. Such issues often emerge repeatedly and are worthy of attention regardless of IUO [21]. Conversely, parents might find it more plausible or intuitive to attribute their child’s severe IUO to “hard” medical or biological factors like pain. Psycho-social aspects, being less tangible, could lead parents to struggle with understanding the influence these may have on their child’s condition.

Despite the observed differences in judgment and the overall weak to moderate agreement between the various two-party constellations, group comparisons revealed no significant differences between the relevance ratings provided by the four rating parties. This circumstance may be due to the small sample size of 22 children. Future studies should explore this research question with a larger sample size. However, if the lack of significant differences is consistently replicated, it would indicate that in a pediatric palliative care inpatient setting, both professionals and parents tend to generate similar hypotheses regarding possible causes of IUO. This is an important basis for establishing a common consensus on which factors are truly relevant and should be treated. This finding is encouraging. Nevertheless, it underscores the importance of ongoing communication and dialogue among the different stakeholders involved in the care of critically ill children to address their individual impressions and any inter-individual discrepancies.

### Limitations

As with the authors’ first study on IUO, the relatively small sample size can be considered the primary limitation of this study. This is particularly evident in the relevance ratings, where the unique characteristics of individual children significantly influence results. Consequently, the generalizability of the average values and findings to all children with IUO may be limited. Nevertheless, this study marks the first successful attempt to systematically evaluate the factors contributing to IUO. Follow-up studies employing a similar approach may yield further important insights into IUO that optimize the long-term treatment of this multifaceted symptom complex.

Due to the study’s design, subjective bias cannot be completely excluded. For example, it was not comprehensively assessed for each PCF whether or not each rater understood it to mean the same construct as another. Completely divergent interpretations are nevertheless unlikely because the initial identification of PCFs occurred during routine clinical practice through a structured diagnostic and therapeutic approach in which professional raters, in particular, have a similar basic understanding of PCFs. Similarly, parents acquire a tremendous amount of expertise over the years and understand what the PCFs are, especially towards the medical PCFs, which often offer little room for interpretation (e.g., seizure). The psycho-social PCFs were also frequently identified through dialogue between parents and professionals, where the alignment of mutual understanding is likely.

Another limitation to consider is that only one rater from each profession was interviewed. In theory, it is possible that another person from the same profession might have a different subjective impression of the individual child compared to the one interviewed. However, this challenge can only be addressed to a certain extent, as the number of individuals involved in a child’s care is limited. In the future, efforts could be made to enhance the standardization of the raters’ interactions with a child to mitigate this limitation.

## 5. Conclusions

In pediatric palliative care, particularly in the inpatient setting, numerous hypotheses from various healthcare professionals and parents emerge during diagnostics as they seek to identify the causes of a child’s IUO. This compilation is crucial for generating an overview of potential therapeutic approaches. However, over time, some factors may turn out to be irrelevant to IUO, while others become extremely important.

The possible origins of IUO can be identified using a standardized diagnostic and therapeutic approach that incorporates a testimony from involved healthcare practitioners. It is important to acknowledge that differences among individual opinions may exist. Nevertheless, experts in pediatric palliative care, which necessarily include parents, may collectively develop a common understanding of IUO.

## Figures and Tables

**Figure 1 children-10-01726-f001:**
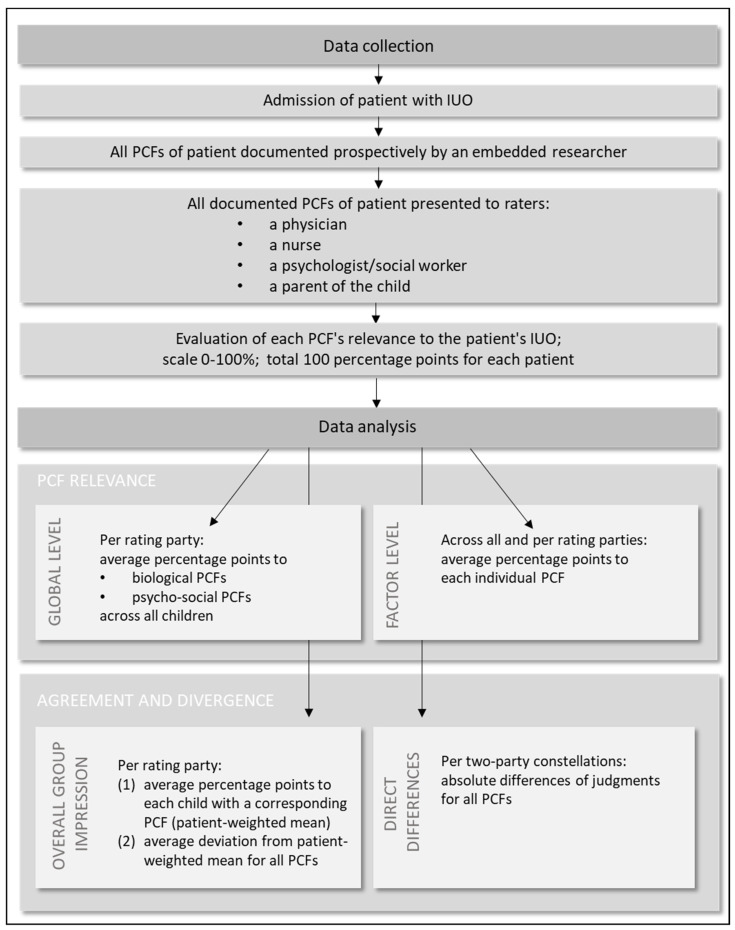
Schematic representation of the methodology chosen to determine PCF relevance and the agreement/divergence of the four rating parties.

**Figure 2 children-10-01726-f002:**
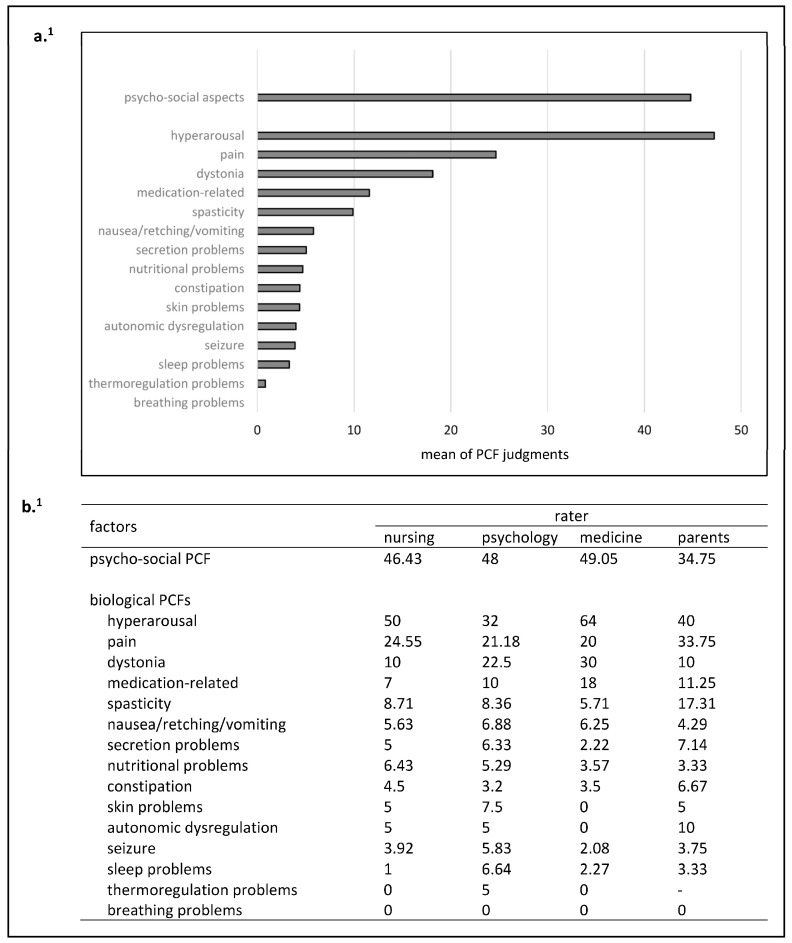
Overall mean of all judgements on a PCF (**a**) and mean ratings of the four parties for the individual PCFs (**b**). ^1^ Averaged over all (party-specific) ratings for a particular PCF.

**Figure 3 children-10-01726-f003:**
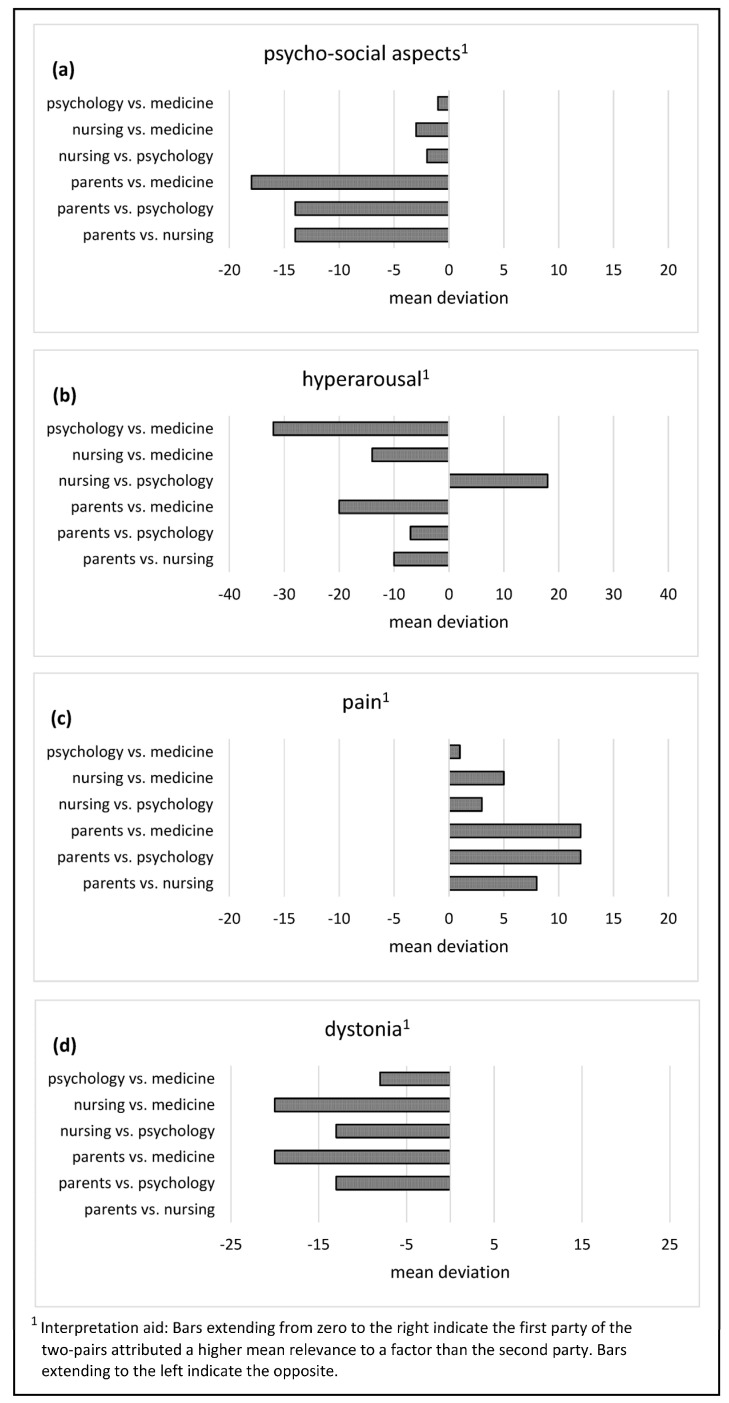
Mean deviations between the judgments of each of two of the four rating parties for the psycho-social PCFs (**a**) and the three most relevant biological factors hyperarousal (**b**), pain (**c**) and dystonia (**d**).

**Table 1 children-10-01726-t001:** Identified categories of potential contributing factors (PCFs) that were presented to professionals and parents for weighting and assessing relevance to IUO ^1^.

Number	Category
1	pain
2	psycho-social aspects
3	spasticity
4	seizure
5	sleep problems
6	constipation
7	secretion problem
8	nausea/retching/vomiting
9	nutritional problems
10	medication-related
11	hyperarousal
12	breathing problems
13	autonomic dysregulation
14	dystonia
15	skin problems
16	thermoregulation problems

^1^ Raters were presented with only those PCFs identified for each child.

**Table 2 children-10-01726-t002:** Average deviations of the four rating parties from the average patient-weighted PCF relevance ratings.

PCFs	Rater			
	Nursing	Psycho-Social	Medicine	Parents
psycho-social PCFs	10.79	7.85	38.89	28.57
biological PCFs				
hyperarousal	13.50	17.83	50.50	32.50
pain	8.91	7.91	17.91	28.14
dystonia	8.13	6.88	21.88	4.38
medication-related	9.08	6.92	10.58	9.58
spasticity	5.63	5.51	4.80	13.92
nausea/retching/vomiting	3.18	3.07	5.36	5.54
secretion problems	2.88	2.90	2.51	6.46
nutritional problems	2.81	2.12	2.45	4.00
constipation	3.20	2.60	2.45	4.50
skin problems	0.63	3.13	0.63	1.88
autonomic dysregulation	1.25	1.25	1.25	0
seizure	2.73	3.19	2.73	4.66
sleep problems	2.64	4.09	3.32	3.11
thermoregulation problems	0.83	1.67	0.83	-
breathing problems	0	0	0	0

**Table 3 children-10-01726-t003:** Agreement between the six rating two-party constellations regarding judgments of all PCFs as measured via Krippendorff’s alpha.

		Krippendorff’s Alpha		
Pairing		α	95%-Confidence Interval (CI)	
Rater 1	Rater 2		Lower	Upper
nursing	parents	0.69	0.58	0.78
medicine	parents	0.64	0.50	0.76
psychology	parents	0.62	0.48	0.74
nursing	psychology	0.61	0.48	0.73
nursing	medicine	0.70	0.57	0.81
psychology	medicine	0.64	0.51	0.75

## Data Availability

The datasets underlying this study are available from the corresponding author upon reasonable request.

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
