# Peer review of "Relevance of Potential Contributing Factors for the Development and Maintenance of Irritability of Unknown Origin in Pediatric Palliative Care"

_children, 2023, doi:10.3390/children10111726_

Round 1

Reviewer 1 Report

Dear colleagues

Thank you for submitting the manuscript entitled: Irritability of Unknown Origin in pediatric palliative care : potential contributing factors’ role to the Children Journal. I read the manuscript and I think you should consider the following points about your manuscript.

1.     Title: It is better to change the title of your manuscript to show that your work is different from Dreier LA, Angenendt N, Hasan C, Zernikow B, Wager J. Potential Contributing Factors for Irritability of Unknown Origin in Pediatric Palliative Care. J Pain Symptom Manage. 2022 Aug;64(2):156-167. doi: 10.1016/j.jpainsymman.2022.04.168. Epub 2022 Apr 14. PMID: 35430284.

2.     Abstract: The method of the abstract should be a brief paragraph of the primary Method of the manuscript. Please correct it. Furthermore, the conclusion in the abstract is not clear.

3.     Introduction: Please write the introduction in three consecutive paragraphs.

4.     Material and Methods: The 2nd paragraph is not related to method. How did you assess the pain score in these patients? Did you use a special tool? What do you mean by skin problem? Please clarify this and other factors in Table 1.

5.     Discussion: In the first paragraph of the discussion you should write the findings of your study clearly. According to Ref 7, what is new in this study?

 Best regards.

Dear colleagues

It is better to write this manuscript according to medical journalism style and native English editing.

Best regards

Author Response

children-2612445

Answers to Reviewer 1

Dear colleagues

Thank you for submitting the manuscript entitled: Irritability of Unknown Origin in pediatric palliative care : potential contributing factors’ role to the Children Journal. I read the manuscript and I think you should consider the following points about your manuscript.

Thank you for the opportunity to submit a revision for our manuscript.

  1. Title: It is better to change the title of your manuscript to show that your work is different from Dreier LA, Angenendt N, Hasan C, Zernikow B, Wager J. Potential Contributing Factors for Irritability of Unknown Origin in Pediatric Palliative Care. J Pain Symptom Manage. 2022 Aug;64(2):156-167. doi: 10.1016/j.jpainsymman.2022.04.168. Epub 2022 Apr 14. PMID: 35430284.

Thank you for this suggestion. As the new title, we recommend "Relevance of potential contributing factors to the development and maintenance of irritability of unknown cause in pediatric palliative care."

  1. Abstract: The method of the abstract should be a brief paragraph of the primary Method of the manuscript. Please correct it. Furthermore, the conclusion in the abstract is not clear.

We have clarified the chosen methodology of our work and revised the Conclusion:

- “Assessments of nursing, psycho-social and medicine professional and parents were used to determine the contribution of different PCFs in the development and maintenance of IUO. For this, individual PCFs of N=22 inpatient children with IUO were presented to the four raters.”

- Understanding which psychosocial and various biological PCFs are significant for IUO can contribute to more targeted and individualized pediatric palliative care in affected patients.

  1. Introduction: Please write the introduction in three consecutive paragraphs.

We have revised the layout of the Introduction and formed three paragraphs.

  1. Material and Methods: The 2ndparagraph is not related to method. How did you assess the pain score in these patients? Did you use a special tool? What do you mean by skin problem? Please clarify this and other factors in Table 1.

Thank you for this comment. We have included additional information to make the methodology clearer:

- p. 3, ll. 104-107: “The diagnostic and therapeutic (standard) approach in the pediatric palliative care unit includes comprehensive basic diagnostics (e.g., physical examination, patient history), special diagnostics (e.g., radiologic/sonographic examinations), and hypothesis-guided therapy (e.g., medication treatment for suspected visceral hyperalgesia). ”

  1. Discussion: In the first paragraph of the discussion you should write the findings of your study clearly. According to Ref 7, what is new in this study?

We have revised the Discussion to more clearly emphasize the differences between Part 1 and this Part 2 of the study in the first paragraph:

- p. 9, ll. 223-228: “Part 1 of this study, using a standardized diagnostic approach, demonstrated that numerous PCFs can potentially account for the development and maintenance of IUO in pediatric palliative care (7). This Part 2 sought, based on the judgment of various professionals and parents, to evaluate and quantify this set of PCFs in terms of their actual relevance to IUO. Psycho-social aspects, hyperarousal, pain and dystonia were considered the most relevant PCFs for IUO.”

Reviewer 2 Report

Dear authors:

Congratulations on your work.

Please review the presented abstract. It should contain: abstract, aim, methodology, main results and main conclusions.

In conclusion please try to explain why your work is relevant in pediatric palliative care? What is the expected impact of your findings?

Best regards.

Minor editing of English language required

Author Response

children-2612445

Answers to Reviewer 1

Dear authors:

Congratulations on your work.

Thank you for the opportunity to submit a revision for our manuscript.

Please review the presented abstract. It should contain: abstract, aim, methodology, main results and main conclusions.

In conclusion please try to explain why your work is relevant in pediatric palliative care? What is the expected impact of your findings?

Thank you very much for this comment. We have revised the abstract and in particular emphasized the chosen methodology more clearly. We have emphasized the relevance of our work to pediatric palliative care. We have nevertheless omitted subheadings in the abstract in accordance with journal requirements for authors:

- “Assessments of nursing, psycho-social and medicine professional and parents were used to determine the contribution of different PCFs in the development and maintenance of IUO. For this, individual PCFs of N=22 inpatient children with IUO were presented to the four raters.”

- Understanding which psychosocial and various biological PCFs are significant for IUO can contribute to more targeted and individualized pediatric palliative care in affected patients.

Reviewer 3 Report

Thanks for letting me review this manuscript. 

I have read the paper of the mother study, and I find it interesting. However, I have doubts that this secondary analysis can be of interest to the reader. Although the commitment and rigorousness of the authors is very clear my opinion is that the topic does not offer results with relevant implications for research and clinical practice. 

There are also a few critical points that I think hinder the publication.

First, it does not seem that the variables associated with irritability have been theoretically and empirically chosen. It is important that potential determinants be scientifically selected

Second, how the assessment were done by the four groups of individuals, is not reported. Consequently, important problems of reliability and validity may arise, which hinders the scientific soundness of the paper

Third, the rationale for the aim of the study is not sustained adequately. What is the gap of the literature? How the results can be used in further research and practice?

I wish all authors well on their studies

moderate proofreading required

Author Response

children-2612445

Answers to Reviewer 3

Thanks for letting me review this manuscript. 

I have read the paper of the mother study, and I find it interesting. However, I have doubts that this secondary analysis can be of interest to the reader. Although the commitment and rigorousness of the authors is very clear my opinion is that the topic does not offer results with relevant implications for research and clinical practice. 

Thank you for your consideration and acknowledgement of our efforts. We respect that you consider the findings of this paper to be too little relevant for practice. Nevertheless, we would like to disagree with this: The diagnosis of IUO in pediatric palliative care is highly complex and it is the lived reality of practitioners that a multitude of possible causes for this distressing symptom come into question (Part 1 of our paper). However, until now, there has never been an attempt to systematically quantify this large set of factors and have them assessed by those who have a good overview of an affected child. Our second part can provide guidance to practitioners as to which factors should be given special attention for the assessment of IUO and can thus a) contribute to a more efficient and economical diagnostic procedure, b) spare the vulnerable patients great burden of long and poorly targeted diagnostics, and c) lead more quickly to adequate therapeutic action and thus higher quality of life for the patients.

There are also a few critical points that I think hinder the publication.

First, it does not seem that the variables associated with irritability have been theoretically and empirically chosen. It is important that potential determinants be scientifically selected.

We have more clearly emphasized in the methodology that the potential contributing factors (PCFs) for IUO were identified using a diagnostic and therapeutic approach that is theoretically, practically, and empirically sound:

  • 3, ll. 107-110: “The approach reflects many years of profound clinical experience and expertise of different professions (e.g. nursing, psycho-social, medical) and the current state of scientific knowledge on IUO in children and adolescents with life-limiting conditions.”

Second, how the assessment were done by the four groups of individuals, is not reported.

We have emphasized this aspect more clearly:

  • 3, ll. 11-121: “For this second part of the study, at the end of the inpatient stay, all documented PCFs of a child were presented individually by a member of the study team to one professional from each of the professions of medicine, nursing, and psycho-social working in the pediatric palliative care unit and caring for the respective child, as well as to one par-ent of the child. Table 1 shows all 16 PCFs identified and classified in the first part of the study that were presented to the raters (7). These raters were then asked to give their as-assessment of how relevant they would rate each PCF for the child’s IUO that has been ad-dressed during the current in-patient care (based on their personal well-grounded expertise towards that specific child).. For this weighting, a total of 100 percentage points could be assigned across all PCFs […]”

Consequently, important problems of reliability and validity may arise, which hinders the scientific soundness of the paper

Thank you for this comment. We hope that by addressing your previous points, we were able to improve your impression of the reliability and validity of the study, which we find to be given.

Third, the rationale for the aim of the study is not sustained adequately. What is the gap of the literature? How the results can be used in further research and practice?

We have emphasized this aspect more clearly in the Abstract and Discussion:

  • Abstract: “Understanding which psychosocial and various biological PCFs are significant for IUO can contribute to more targeted and individualized pediatric palliative care in affected patients.”
  • - 9, ll. 223-228: “Part 1 of this study, using a standardized diagnostic approach, demonstrated that numerous PCFs can potentially account for the development and maintenance of IUO in pediatric palliative care (7). This Part 2 sought, based on the judgment of various professionals and parents, to evaluate and quantify this set of PCFs in terms of their actual relevance to IUO. Psycho-social aspects, hyperarousal, pain and dystonia were considered the most relevant PCFs for IUO.”

I wish all authors well on their studies

Round 2

Reviewer 1 Report

Dear colleagues 

Thank you for revising your manuscript, right now it is more acceptable.

Best regards

Dear colleagues

Thank you again, in my opinion, it is better to do native editing.

Best regards

Author Response

Thank you for this positive feedback. Of course, we have complied with your request to have the manuscript checked by a native speaker. Merely, the feedback was not yet available at the submission deadline of the last revision round. The current version of the manuscript corresponds to the version corrected by the native speaker. Of course, the native speaker did not make any changes to the content of the manuscript.

Reviewer 3 Report

Thanks for the response of the authors.

I still believe that this type of study is more qualitative than quantitative, because there is lack of validity and reliability of the measures taken into account. For example, how do the authors know that the parents judge the factors by attributing the same meaning of the symptoms that the nurses and psychologists do? Other comparisons may still apply. Therefore, generalization is very limited.

This reviewer claimed that the results are not particularly impacting, given these problems in measurements.

I understand that this work may be important in the specific context. therefore, I think that the article should be integrated with a comprehensive limitations paragraphs, before it can be published

The other points have been adequately addressed thanks.

minor editing

Author Response

Thank you very much. We have included this important aspect in the limitations of the study:

Due to the study design, subjective bias cannot be completely excluded. For example, it was not comprehensively assessed for each PCF whether each rater understood it to mean the same construct. Completely divergent interpretations are nevertheless unlikely because the initial identification of PCFs occurred during routine clinical practice through a structured diagnostic and therapeutic approach in which professional raters, in particular, have a similar basic understanding of PCFs. Similarly, parents acquire a tremendous amount of expertise over the years and understand what the PCFs are, especially towards the medical PCFs, which often offer little room for interpretation (e.g., seizure). The psycho-social PCFs were also frequently identified through dialogue between parents and professionals, where alignment of mutual understanding is likely.
